# Novel Long Non-Coding RNA (lncRNA) Transcript AL137782.1 Promotes the Migration of Normal Lung Epithelial Cells through Positively Regulating LMO7

**DOI:** 10.3390/ijms241813904

**Published:** 2023-09-09

**Authors:** Ying Zhang, Weili Wang, Chunchun Duan, Min Li, Liyang Gao

**Affiliations:** 1Life Science School, Ningxia University, Yinchuan 750001, China; zhangying191231@126.com (Y.Z.); w15709104535@163.com (W.W.); d18793811649@aliyun.com (C.D.); 2Key Laboratory of Ministry of Education for Conservation and Utilization of Special Biological Resources in the Western, Ningxia University, Yinchuan 750001, China

**Keywords:** lncRNA AL137782.1, lung epithelial cells, cell migration, LMO7

## Abstract

The role of long non-coding RNA (lncRNAs) in biological processes remains poorly understood, despite their significant impact. Our previous research discovered that the expression of AL137782.1, a long transcript of the novel lncRNA ENSG00000261553, is upregulated in lung epithelial cells upon exposure to microbes. Furthermore, the expression of AL137782.1 exhibits variability between para-cancerous and lung adenocarcinoma samples. These findings imply that this lncRNA may play a role in both normal lung epithelial cellular processes and pathophysiology. To elucidate the function of AL137782.1 in lung epithelial cells, we utilized bioinformatics retrieval and analysis to examine its expression. We then analyzed its subcellular localization using fluorescence in situ hybridization (FISH) and subcellular fractionation. Through rapid amplification of cDNA ends (RACE), we confirmed the presence of a 4401 nt lncRNA AL137782.1 in lung epithelial cells. Moreover, we discovered that this lncRNA positively regulates both mRNA and the protein expression of LMO7, a protein that may regulate the cell migration of normal lung epithelial cells. Although the overexpression of AL137782.1 has been shown to enhance the migration of both normal lung epithelial cells and lung adenocarcinoma cells in vitro, our study revealed that the expression of this lncRNA was significantly decreased in lung cancers compared to adjacent tissues. This suggests that the cell migration pattern regulated by the AL137782.1–LMO7 axis is more likely to occur in normal lung epithelial cells, rather than being a pathway that promotes lung cancer cell migration. Therefore, our study provides new insights into the mechanism underlying cell migration in human lung epithelial cells. This finding may offer a potential strategy to enhance normal lung epithelial cell migration after lung injury.

## 1. Introduction

The lung epithelium is a crucial physical barrier that protects the lung from environmental aggressors, stabilizing lung function and resisting invasion by foreign pathogens [1]. It consists of bronchial and alveolar epithelial cells, as well as pulmonary endothelial cells, which together form a complex internal structure [2]. Bronchial epithelial cells play an essential role in airway defense by providing mucosal, ciliary, and mechanical barriers. Alveolar epithelial cells exhibit strong plasticity and self-renewal abilities, crucial for lung development and repair [1]. Studies have indicated that alveolar type Ⅱ epithelial cells possess the ability to restore the normal alveolar structure through migration and proliferation during an injury [3,4]. Additionally, alveolar type Ⅱ epithelial cells have the ability to differentiate into alveolar type I epithelial cells, which cover abnormal pulmonary surfaces and maintain the integrity of the pulmonary epithelial barrier [5]. Hence, comprehending the migration mechanism of lung epithelial cells would establish a fundamental basis for promoting the repair of lung injuries and offer valuable insights for the treatment of lung-related diseases.

Long non-coding RNA (lncRNAs) are RNA molecules with a length of over 200 nucleotides (nt) [6]. Despite being originally thought to be incapable of encoding proteins [7], recent reports suggest that some lncRNAs contain open reading frames that do in fact encode peptides [8,9]. More evidence has confirmed that lncRNAs play important roles in various cellular processes, such as cell proliferation, migration, differentiation, and metabolism [10,11,12]. Due to their diversified localization, complex structure and tissue specificity, the biological function of lncRNAs has not been fully elucidated up to now [13,14]. With the development of science and technology, high-throughput sequencing and bioinformatics methods have made great progress in lncRNAs identification. In our previous research, we identified AL137782.1 as a novel differentially expressed lncRNA in lung epithelial cells after infection with *Mycoplasma ovipneumoniae* [15]. However, the subcellular localization and biological function of AL137782.1 in lung epithelial cells remain unclear, and the mechanism by which it regulates lung epithelial cell migration, as well as its potential as a target for lung repair, has not been reported. Therefore, investigating the role and mechanism of AL137782.1 in lung epithelial cell migration provides new perspectives and ideas for lung injury repair research.

## 2. Results

### 2.1. LncRNA AL137782.1 Highly Expressed in Para-Cancerous Samples Than Lung Adenocarcinoma

Using the TCGA database and UALCAN database, we discovered that the expression of AL137782.1 is observed in various types of cancers, including the two main types of lung cancer, namely lung adenocarcinoma (LUAD) and lung squamous cell carcinoma (LUSC) [16,17]. Bioinformatics analysis revealed that the expression of the AL137782.1 transcript was lower in primary LUAD samples (533) compared to normal samples (59) (*p* = 0.00719, Figure 1A). Similarly, the expression of AL137782.1 in primary LUSC samples (502) was lower compared to normal samples (49) (*p* = 0.0031, Figure 1B). Although the expression level of AL137782.1 did not influence the overall survival of LUAD patients (Figure 1C), it was found to be significantly associated with the survival rate of LUSC patients (Figure 1D).

### 2.2. The Profile of lncRNA AL137782.1 in Human Bronchial Epithelial Cells and Type II Alveolar Epithelial Cell

According to the Ensembl and UCSC databases, AL137782.1 (ENSG00000261553) is located on human chromosome 13 (Figure 2A). AL137782.1 has two transcripts, AL137782.1-201 (ENST00000563635.5) and AL137782.1-202 (ENST00000570285.1), and their biotypes are lncRNA. AL137782.1-201 is relatively long, its genome location is similar to that of UCHL3 and LMO7, and its exons partially overlap with LMO7 (Appendix A). Therefore, the function of AL137782.1 transcript 201 was further examined in this study. In order to identify the coding potential of AL137782.1, ORF Finder (https://www.ncbi.nlm.nih.gov/orffinder/, accessed on 7 September 2023) was used to analyze the open reading frames (ORFs) of AL137782 (Figure 2B), and the results show that AL137782.1 has 33 ORFs, which encode short peptides composed of 28–302 amino acids, respectively. These short peptides have no homologous protein, indicating that AL137782.1 might cannot encode a protein (Appendix A). Furthermore, the coding potential of AL137782.1 was analyzed using *CPAT* (Figure 2C), and the results show that the coding potential of AL137782 is low (only 0.07) and there was no coding label. In summary, AL137782.1 reflects the characteristics of long non-coding RNA and belongs to the lncRNA, which cannot code protein.

To determine the full length of AL137782.1, we performed 5′ and 3′ rapid amplification of complementary DNA ends (RACE). The RACE result showed that the 5′ end of AL137782.1 was 273 nt; however, no amplification of the 3′ end was detected, suggesting that AL137782.1 might not have a 3′ poly(A) structure. The length of AL137782.1 was reported to be 4401 nt in lung epithelial cells (Figure 2D,E, Appendix A). Then, the MFE secondary structure of AL137782.1 in lung epithelial cells was predicted by means of RNAfold, using default parameters. This lncRNA secondary structure model was represented by a 4401 nt sequence of nucleotide pairing states, and colored using base-pairing probabilities (Figure 2F).

### 2.3. AL137782.1 Is Located Both in Nucleus and Cytoplasm of Lung Epithelial Cells

The subcellular localization of lncRNA is closely associated with its biological effects and potential molecular roles. Therefore, we utilized online software and the RNA-FISH assay to detect the subcellular distribution of AL137782.1 in lung epithelial cells. The results indicate that AL137782.1 is predominantly localized in the nucleus and cytoplasm of lung epithelial cells, as observed by the abundant punctate patterns in these compartments (Figure 3A,B, Appendix A). To further validate these findings, overexpression of AL137782.1 was conducted, followed by the isolation of nuclear RNA and cytoplasmic RNA from lung epithelial cells. Using U6 and GAPDH as internal reference genes, we detected the relative expression of AL137782.1 using RT-qPCR (Figure 3C,D), and consistent with the results obtained from RNA-FISH, similar patterns of the subcellular distribution of AL137782.1 were observed.

### 2.4. AL137782.1 Is Co-Expressed with the mRNA Expression of LMO7 in Lung Epithelial Cells

To analyze the function of AL137782.1, control cell lines and AL137782.1 overexpression cell lines were established. Then, total RNA was extracted for transcriptome sequencing. The differently expressed genes (DEGs) between the empty vector control group and AL137782.1 overexpression group were identified, and this result showed that 322 genes were significantly changed, including 101 significantly up-regulated genes and 221 significantly down-regulated genes. The DEGs in the control and treated groups are presented using a volcano plot in Figure 4A. To further clarify the potential biological functions of AL137782.1 in lung epithelial cells, the DEGs and transcripts were analyzed by means of GO and KEGG enrichment analysis (Figure 4B). In particular, these mRNAs were primarily associated with the regulation of type I interferon, the regulation of virus life processes, immune response and the secretion of cytokines. In addition, 20 KEGG pathways were enriched, including 10 significantly up-regulated signaling pathways and 10 significantly down-regulated signaling pathways (Figure 4C). Among these enriched pathways were those closely related to host defense, adhesion junctions and immune regulation.

To further explore the interaction between DEGs and the AL137782.1 lncRNA transcript, weighted gene co-expression network analysis (WGCNA) was conducted. The co-expression relationship network of lncRNA and mRNA showed that the expression levels of LMO7 and AL137782.1 are consistently linearly correlated (Figure 4D). It is suggested that *LMO7* is solely related to AL137782.1 and highly co-expressed with AL137782.1.

To target the relationship between AL137782.1 and the expression of LMO7 mRNA and protein, two AL137782.1 overexpression cell lines, the AL137782.1-overexpressed A549 cell line and the AL137782.1-overexpressed BEAS-2B cell line, were established. The mRNA expression of *LMO7* was analyzed using RT-qPCR, while the protein expression was assessed using Western blot and immunofluorescence. The RT-qPCR results showed that mRNA expression of *LMO7* was significantly up-regulated (Figure 5A), while the Western blot results demonstrated a significant upregulation of LMO7 protein expression in A549 and BEAS-2B cells with AL137782.1 overexpression (Figure 5B). We also observed an accumulation of LMO7 protein (shown by red fluorescence) in AL137782.1-overexpressing cell lines, as demonstrated by means of immunofluorescence analysis (Figure 5C,D). Furthermore, two small interfering RNAs (si-AL137782.1-#1 and si-AL137782.1-#2) were used to knock down the expression of AL137782.1 in A549 and BEAS-2B cells. Compared with si-AL137782.1-#1, the expression of AL137782.1 was significantly reduced in A549 and BEAS-2B cells following transfection with si-AL137782.1-#2 (Figure 6A,B). The *LMO7* mRNA level was decreased after transfection (*p* < 0.05, Figure 6C); this finding also verified via Western blot analysis (Figure 6D,E) and immunofluorescence (Figure 6F,G). These findings above confirm that AL137782.1 positively regulates the expression of both LMO7 mRNA and protein, indicating a potential regulatory role of AL137782.1 in A549 and BEAS-2B cells.

To confirm AL137782.1 positively regulates the expression of *LMO7*, we examined the expression of *LMO7* in LUAD and LUSC as well (TCGA database and UALCAN database). Similar to the expression level of AL137782.1 in LUAD and LUSC, the expression level of *LMO7* was higher in normal samples (Figure 7A,B).

### 2.5. AL137782.1 Regulates the Migration of Lung Epithelial Cells though LMO7

Next, we aimed to elucidate the potential impact of AL137782.1 in lung epithelial cells. A wound healing assay was performed to evaluate the migration ability of lung epithelial cells. This result showed that the wound healing rate of the AL137782.1 overexpression groups was significantly accelerated (Figure 8A,B), indicating that AL137782.1 could promote the migration of lung epithelial cells. Meanwhile, siRNA si-AL137782.1-#2 was used to inhibit the expression of AL137782.1 in lung epithelial cells, and the wound healing assay showed that the cell migration rate was inhibited (Figure 8C,D).

To further explore whether AL137782.1 affects the cells migration via LMO7, LMO7 was knocked down in AL137782.1 overexpression cells (Figure 9A,B), following by the wound healing assay. We found that the cell migration rate of AL137782.1 overexpression cells significantly decreased with a lower LMO7 expression (Figure 9C,D). These results above provide evidence that AL137782.1 positively regulates the cell migration of lung epithelial cells though the expression of LMO7.

## 3. Discussion

Our previous work has reported that the expression of lncRNA AL137782.1 in lung epithelial cells was related to infection with microbes [15]. lncRNA AL137782.1 has also been reported in colon cancer, and it has been found to relate to m6A modification. However, the biological function of lncRNA AL137782.1 in lung epithelial cells remains unknown [18]. Therefore, in this study, lncRNA AL137782.1 was examined at three levels: the profile of AL137782.1, the biological function of AL137782.1 *in vitro*, and its molecular mechanisms.

Firstly, this transcript has been found in multiple lung epithelial cell lines, including BEAS-2B, A549, and HPAEpiC. It also showed a negative expression in two lung tumors, LUAD and LUSC, which originate from lung epithelial cells. These findings indicate that AL137782.1 regulates the cellular process of normal lung epithelial cells rather than tumor cells.

Secondly, the profile of a novel lncRNA transcript AL137782.1 has been demonstrated. Although sequencing the whole genome transcripts of mammalian cells, tissues or serum samples constitutes the basis for screening lncRNA, a series of classic molecular biology techniques still need to be used for the experimental verification of lncRNA. The 5′ RACE/3′ RACE experiment was used to determine the full length of mRNA, such as the start site and termination site of transcription [19,20,21]. We found that the length of AL137782.1 is 4410 nt in lung epithelial cells and the 5′ end of AL137782.1 was 273 nt long. Then, after we used this sequence as target for the structure of this lncRNA, its second structure indicated that AL137782.1 might have three main parts, namely a heavily forked head, a middle trunk, and a heavily forked tail. Then, fluorescence in situ hybridization was used to determine the localization of lncRNAs [22,23], and the expression of AL137782.1 was observed in both nucleus and cytoplasm. The subcellular localization of lncRNAs is often related to their functions in transcriptional regulation or post-transcriptional regulation, and some lncRNAs are distributed in the chromatin, nucleus or cytoplasm [24]. The AL137782.1 transcript being distributed in both the nucleus and cytoplasm of A549/BEAS-2B cells suggested that AL137782.1 might play roles in both transcription and post-transcriptional regulation. Therefore, AL137782.1 might perform multiple functions in lung epithelial cells.

Then, we retrieved the location and exons of AL137782.1 in the genome using the Ensembl and UCSC databases, and found that its genome location was similar to that of LMO7 and UCHL3, and its exons partially overlapped with *LMO7*. Using ORF Finder and CPAT to analyze the sequence of AL137782.1, it was found that AL137782.1 had 33 ORFs, but they were all short peptides. Its coding potential was extremely low, and there was no coding label, which was consistent with the characteristics of lncRNA.

Thirdly, we explored the downstream target gene and pathways of AL137782.1 and its direct functions. Further studies using transcriptome sequencing and bioinformatics analysis revealed that AL137782.1 might relate to host defense, autoimmune disorders, adhesion and immune regulation. Importantly, AL137782.1 was found to positively regulate the expression of LMO7.

LMO7, a member of the PDZ- and LIM domain-containing protein family, has functions involved in myoblast differentiation [25], cells proliferation and migration [26]. LMO7 is also essential for skeletal muscle development and cardiac function [27]. In myoblast cells, the myoblast-differentiation-related genes *Pax3*, *Pax7*, *Myf5* and *MyoD* were significantly decreased after the downregulation of LMO7, while overexpressing LMO7 significantly increased *MyoD* and *Myf5* [27]. Previous studies showed that circulating exosomal microRNA-96 promotes cell proliferation, migration and drug resistance by targeting LMO7 [26]. In the investigation of the mechanisms underlying cell-specific transcription driven by MRTFs and SRF, it has been found that LMO7 acts as a cell-specific regulator of MRTFs. Under the synergistic influence of Rho GTPases, LMO7 might alleviate the inhibitory effect mediated by actin [28]. On the other hand, LMO7 may be involved in protein–protein interactions. According to the yeast two-hybrid screening, LMO7 and LIMCH1 were identified as interaction partners to LRIG3 [29], and it was proven that LMO7 and LIMCH1 physically interact with LRIG proteins and could regulate the process of non-small cell lung cancer, which has prognostic implications for early-stage disease. Currently, several lines of evidence suggest that LRIG1 protein can regulate cell migration [30,31], but it is unclear whether the interaction between LMO7 and LIMCH1 and LRIG protein has an important effect on cell migration. LMO7 is a fibrous actin-binding protein that is widely expressed in adult tissues, particularly at the apical surface of lung epithelial cells [32].

LMO7 expression was up-regulated in certain cancer tissues, such as breast, liver, stomach and lung pancreas cancer. It is considered as a molecule that aids in the formation and maintenance of the epithelial architecture via the remodeling of actin cytoskeleton [33,34]. To clarify the relationship between *LMO7* and AL137782.1 in lung epithelial cells, we overexpressed or knocked down AL137782.1 in A549 and BEAS-2B cells, then examined *LMO7* mRNA and protein expression via RT-qPCR and Western blot. Interestingly, we observed that LMO7 expression positively correlates with AL137782.1 in A549 and BEAS-2B cells. At the same time, we verified that AL137782.1 could promote cell migration without affecting the rate of proliferation (Appendix A). Similarly, by means of loss-of-function studies, we demonstrated that the loss of AL137782.1 resulted in slowed cell migration and decreased LMO7 expression. Further mechanism investigations revealed that AL137782.1 promotes cell migration through increasing LMO7 in A549 and BEAS-2B cells. In addition, the effect of AL137782.1 overexpression on cell migration in A549 and BEAS-2B cells was abolished by means of co-transfection with *LMO7* siRNA. Hence, we argue that AL137782.1 mediates cell migration targeting the LMO7 transcriptional network. Through a comparison of cancer tissue samples and adjacent normal tissue samples, we found that both AL137782.1 and *LMO7* were expressed at higher levels in normal cells than in two types of lung cancers, LUAD and LUSC. This suggests that the cell migration mechanism mediated by the AL137782.1–LMO7 axis is more likely to occur in normal lung epithelial cells, rather than being one of the pathways promoting the migration of lung cancer cells. This study improves our understanding of lncRNA in lung epithelial cells, provides a new idea and reference for the lung epithelial cells research by lncRNA, and enriches the study of lncRNAs in LMO7. Also, the AL137782.1–LMO7 axis may be a therapeutic target for intimal hyperplasia and wound healing.

## 4. Materials and Methods

### 4.1. Cell Cultures

Two human lung epithelial cell lines, A549 and BEAS-2B (Beyotime, Shanghai, China) were cultured in DMEM (Gibco, Grand Island, NY, USA) with 10% FBS (Gibco, Grand Island, NY, USA) and 1% penicillin/streptomycin (Thermo Fisher Scientific Inc., Waltham, MA, USA) in a humidified 5% CO_2_ incubator at 37 °C.

### 4.2. Overexpression of AL137782.1 by Recombinant Adenovirus

AL137782.1 expressing recombinant adenovirus was synthesized in a pH-BAd-EF1-MCS-CMV-EGFP-Δloxp vector by HANBIO (Shanghai, China). To overexpress AL137782.1 in A549 and BEAS-2B cells, cells were infected with adenoviral expression vectors or adenoviral backbone vectors according to the manufacturer’s instructions, and the multiplicity of infection reached 300. After being infected for 8 h, we exchanged the medium for fresh medium with 10% FBS. Thirty-six hours later, RNA was extracted and the overexpression efficiency was determined via real-time PCR.

### 4.3. Rapid-Amplification of cDNA Ends (RACE)

The 5′-RACE and 3′-RACE analyses were used to determine the transcriptional initiation and termination sites of AL137782.1 and were carried out with the SMARTer RACE cDNA amplification kit (Clontech, San Francisco, CA, USA) according to the manufacturer’s instructions. In brief, RNA was extracted from A549 cells overexpressed with AL137782.1. The 3′- and 5′-RACE-ready cDNA was synthesized using the reverse transcriptase contained in this kit. Amplification was performed according to the manufacturer’s instructions. The obtained band was purified and sequenced by Quintara (Wuhan, China). The gene-specific primers used for the RACE assay were as follows:

5′ RACE-1 5′-GATTACGCCAAGCTTTGTCACCGTAGCCACTGTCCCTGCC-3′;

5′ RACE-2 5′-GATTACGCCAAGCTTGTCACCGTAGCCACTGTCCCTGCCA-3′;

3′ RACE-1 5′-GATTACGCCAAGCTT AGTAGTGCTTCGGATGAGCCCAGGTGC-3′;

3′ RACE-2 5′-GATTACGCCAAGCTT GACTCCCATGCAGCCAGGAGAAGCAGT-3′;

Then, the MFE secondary structure of AL137782.1 in lung epithelial cells was predicted via RNAfold using default parameters [35].

### 4.4. Fluorescence In Situ Hybridization (FISH)

The FISH assay was performed in A549 and BEAS-2B cells according to the specifications of the manufacturers. The Cy3-labeled AL137782.1 probe mix used in our study were designed and synthesized by RIBOBIO (Guangzhou, China). Briefly, cells with transduced AL137782.1 were fixed using 4% paraformaldehyde for 30 min. After 0.5% TritonX-100 permeation, the cells were blocked at 37 °C for 30 min with pre-hybridization solution. Then, the cells were incubated with a probe mix at 37 °C overnight. The cell nuclei were stained with DAPI (Beyotime, Shanghai, China). The distribution of AL137782.1 was observed using a confocal laser scanning microscope.

### 4.5. RNA Extraction and Real-Time PCR

Total RNA was extracted from A549 and BEAS-2B cells using the trizol reagent (Beyotime, Shanghai, China) according to the manufacturer’s instructions. Real-time PCR was conducted with the SYBR Green PCR kit (Takara, Kyoto, Japan) to determine the target mRNA expression in this study. The sequences of real-time PCR primers were synthesized by Sangon Biotech (Shanghai, China). Relative mRNA expression was normalized using GAPDH and calculated according to the 2^−ΔΔCt^ method. The sequence of primers were as follows:

AL137782.1 F 5′-TTCCCTTTTTCCTGCTGA TTAC-3′;

AL137782.1 R 5′-GGCCCCTGGTA TCCCTTTCT-3′;

*LMO7* F 5′-GAGCCAAAGACTGCGTTACCCT-3′;

*LMO7* R 5′-TTGCCCAACTTCTTCTGTTA TCCTC-3′.

### 4.6. Subcellular Fractionation

A549 and BEAS-2B cells were lysed on ice with pre-chilled lysis buffer J, and the cell suspension was collected via centrifugation. We inserted the supernatant into a 1.5 mL centrifuge tube without the ribozyme as the cytoplasmic RNA fraction and the pellet as the nuclear RNA fraction. The nuclear and cytoplasmic fractions were isolated with the Cytoplasmic & Nuclear RNA Purification Kit (Norgen, Thorold, ON, Canada) according to the manufacturer’s instructions. Then, we detected the distribution of AL137782.1 in the nucleus and cytoplasm using RT-qPCR.

### 4.7. Transcriptome Sequence

Two groups were established in this experiment, including the control group (adenoviral backbone vectors) and the AL137782.1 overexpression group (adenoviral expression vectors). Then, the total RNA from each group was extracted for transcriptome sequencing (Majorbio, Shanghai, China). The data were analyzed on the online platform of Majorbio Cloud Platform (https://pan.quark.cn/s/1c2206165b90, accessed on 7 September 2023). The criteria for the selection of DEGs were |log2FC| ≥ 1 and *p*-value < 0.05.

### 4.8. Western Blot

A549 and BEAS-2B cells were lysed using Cell lysis buffer for Western and IP (Beyotime, Shanghai, China), and the lysate was subsequently analyzed by Western blotting. The equal amounts of protein were resolved using SDS-PAGE gels, and transferred to PVDF membranes. After blocking with 5% nonfat milk, the membranes were incubated with LMO7 (Novus Biologicals, Littleton, CO, USA) primary antibody at a dilution of 1:1000 overnight at 4 °C, and GAPDH (Proteintech, Wuhan, China) was used as an internal control. Following washing with TBST, the membranes were incubated with HRP-conjugated secondary antibody for 1 h (Beyotime, Shanghai, China) and washing with TBST again, then signals were detected with ECL detection system.

### 4.9. RNA Knockdown by Small Interfering RNA

Small interfering RNAs (siRNAs) targeting AL137782.1 or LMO7 were designed and synthesized by RiboBio (Guangzhou, China), and scrambled oligonucleotides were used as a negative control (NC). In siRNA transfection experiments, A549 and BEAS-2B cells were seeded in six-well plates at a density of 1 × 10^5^/well and incubated overnight. Then, we transfected siRNAs using the riboFECT™CP reagent with a final concentration of 50 nM. Thirty-six hours later, we extracted total RNA and determined interfering efficiency using real-time PCR and Western blot.

### 4.10. Wound Healing Assay

For the wound healing assay, we used 10 μL pipette tips to obtain the cell-free lane. A ruler was used as a guide to obtain a straight line. To remove the detached cells, the culture medium was discarded, and the cells were washed twice with PBS. Cells continued to be cultured and were allowed to migrate for 48 h. The distances between the edges of the scratch were measured to quantitatively evaluate cell migration.

### 4.11. Databases and Bioinformatics Analysis

The clinical data of LUAD and LUSC originated from the TCGA database and assessed using UALCAN (https://ualcan.path.uab.edu/cgi-bin/ualcan-res-lnc.pl, accessed on 7 September 2023; https://ualcan.path.uab.edu/cgi-bin/ualcan-res-lnc.pl, accessed on 7 September 2023), the *p* value of each group was calculated using UALCAN. The MFE secondary structure was generated via RNAfold (http://rna.tbi.univie.ac.at/, accessed on 7 September 2023) using default parameters. R Studio (version 1.1.453) and R Project version 4.0.2 were used for statistical analyses.

### 4.12. Statistical Analysis

Data were collected from at least three independent experiments, with three replicates performed per experiment. Data analysis combined with IBM SPSS Statistics 22 (SPSS Inc., Chicago, IL, USA) and GraphPad Prism 7 (GraphPad Software, San Diego, CA, USA). All data are expressed as the mean ± SD. Comparisons between two groups were conducted using an unpaired *t*-test and those among multiple groups were conducted via one-way ANOVA. *, **, *** represented *p* < 0.05, *p* < 0.01, *p* < 0.001, respectively.

## Figures and Tables

**Figure 1 ijms-24-13904-f001:**
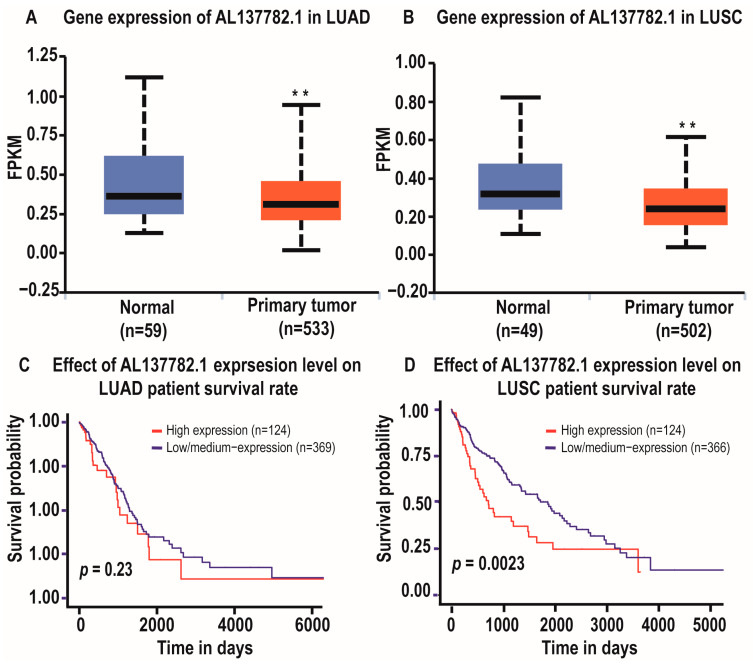
Expression and relative survival rate of lncRNA AL137782.1 in lung cancer groups. (**A**) Expression of AL137782.1 in LUAD; (**B**) expression of AL137782.1 in LUSC; (**C**,**D**) patient survival rate. ** *p* < 0.01.

**Figure 2 ijms-24-13904-f002:**
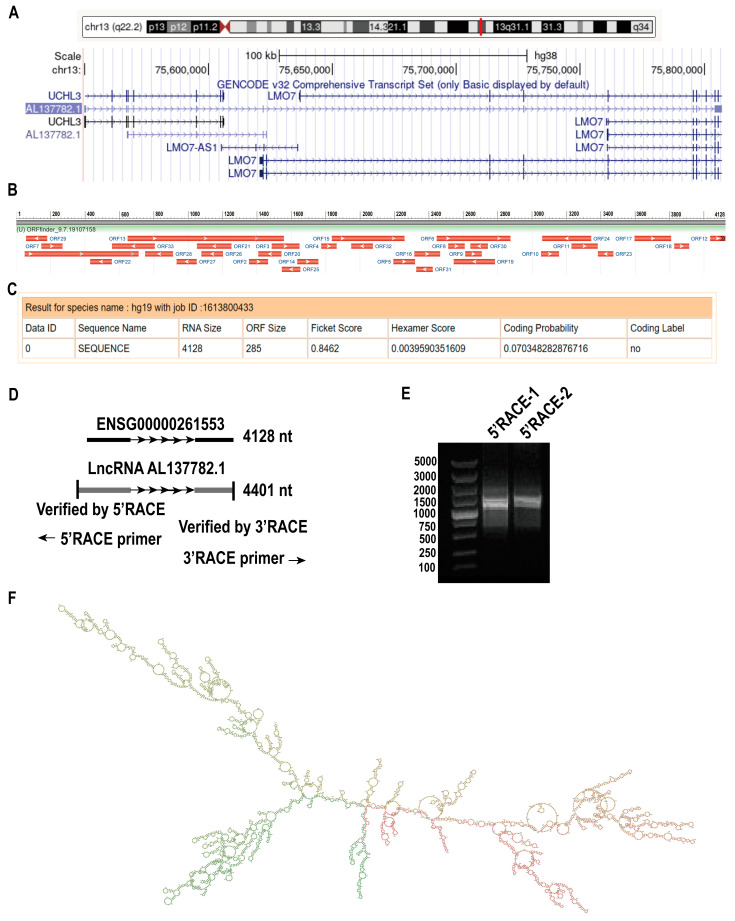
Identification of AL137782.1. (**A**) The position of AL137782.1 on the chromosome; (**B**) the open reading frame of AL137782.1; (**C**) prediction of the encoding capacity of AL137782.1; (**D**) full length of AL137782.1 ENST00000261553 was predicted in the Ensembl database and measured AL137782.1 was acquired via 5′ and 3′ RACE; (**E**) the result of 5′ RACE agarose gel electrophoresis; (**F**) lncRNA secondary structure model.

**Figure 3 ijms-24-13904-f003:**
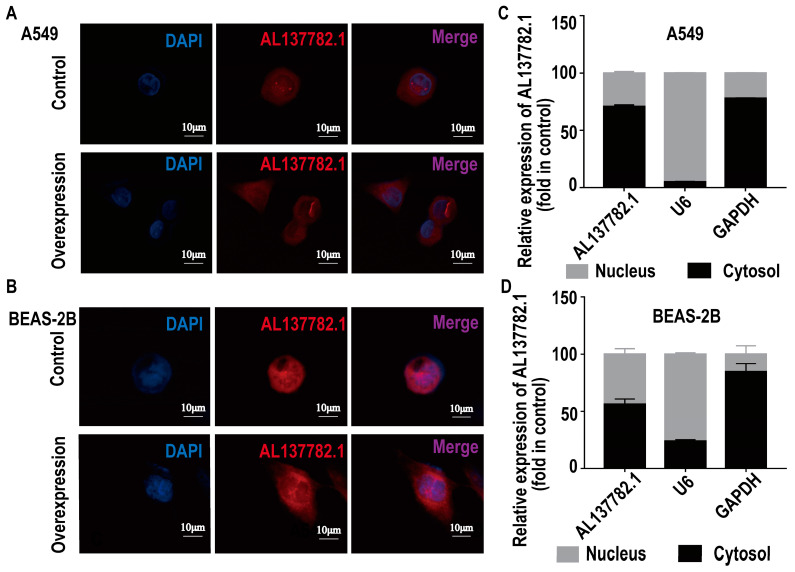
The localization of AL137782.1 in lung epithelial cells. (**A**,**B**) The subcellular localization of AL137782.1 in A549 and BEAS-2B cells was detected via RNA-FISH; (**C**,**D**) RT-qPCR was used to detect the AL137782.1 in the nucleus and cytoplasm of A549 and BEAS-2B cells.

**Figure 4 ijms-24-13904-f004:**
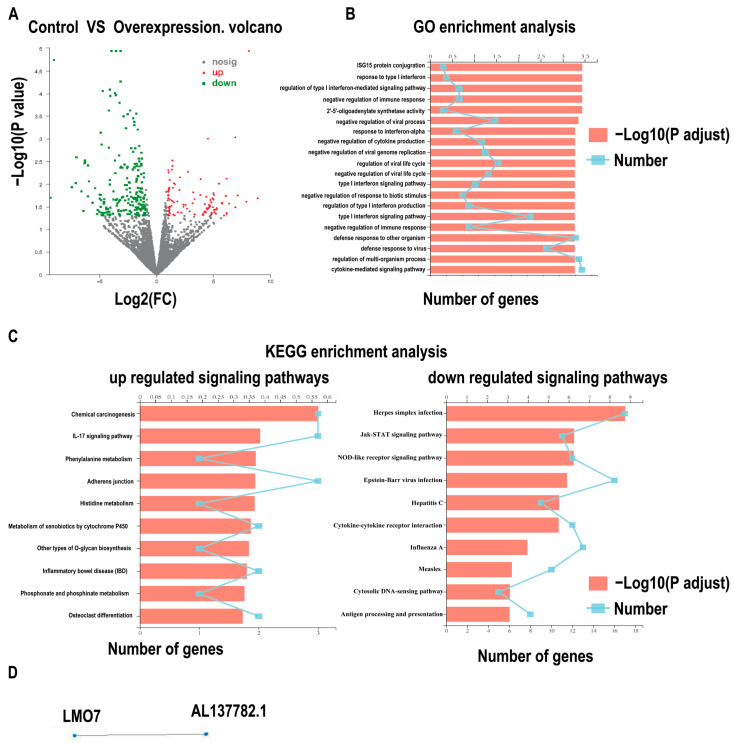
AL137782.1 is solely co-expressed with the mRNA expression of LMO7. (**A**) Volcano plots show the -log10 (*p* value) against log2 (FC) for DEGs in lung epithelial cells, and the green, red, and gray points represent down-expressed, up-expressed, and non-DEGs, respectively (|log2FC| ≥ 1 and *p* value < 0.05), nosig: no significant; (**B**) GO enrichment analysis of DEGs; (**C**) KEGG pathway enrichment analysis of significantly up-regulated and down-regulates expressed genes; (**D**) co-expression network for AL137782.1 and its co-expressed mRNA.

**Figure 5 ijms-24-13904-f005:**
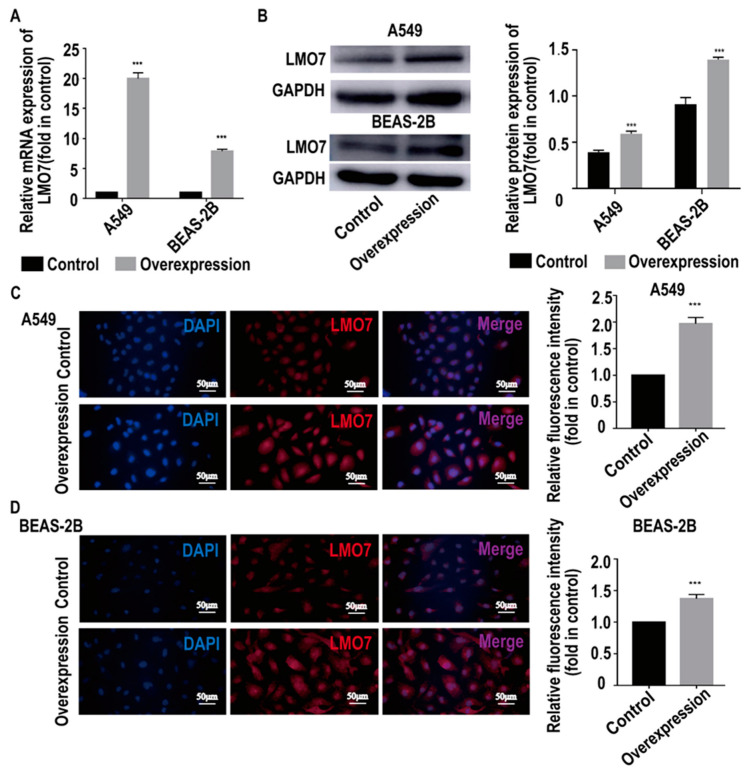
Overexpressing AL137782.1 upregulates the expression of LMO7. (**A**) The mRNA expression of *LMO7* in A549 and BEAS-2B cells were verified by means of RT-qPCR. (**B**) Western blot was used to verify the protein expression of LMO7 in A549 and BEAS-2B cells. (**C**,**D**) Expression of LMO7 in A549 and BEAS-2B cells was determined by means of immunofluorescence. *** *p* < 0.001, ns means no significant difference.

**Figure 6 ijms-24-13904-f006:**
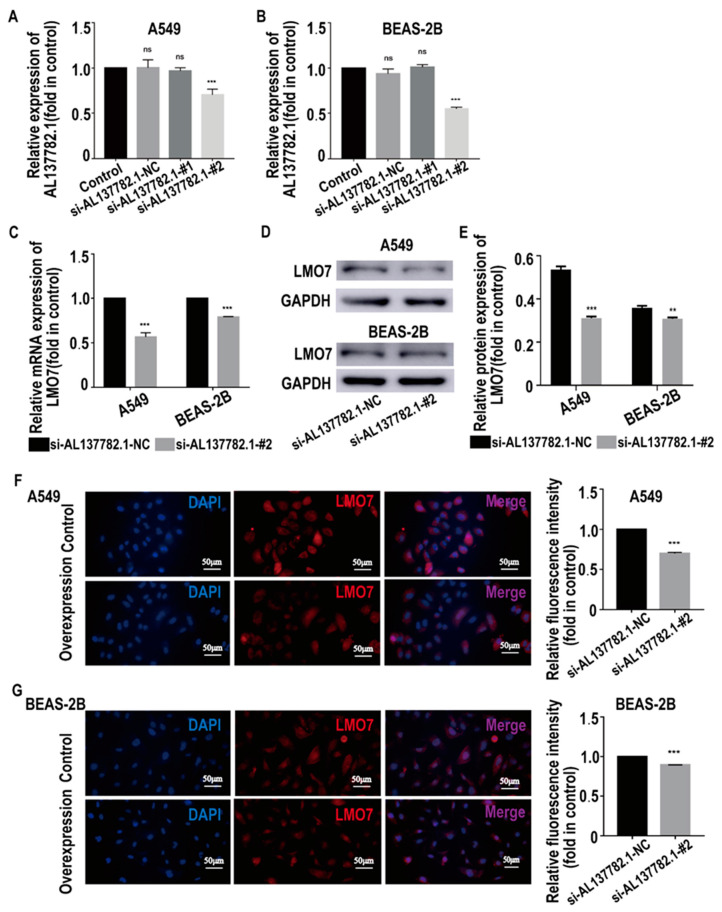
Knocking down AL137782.1 decreases the expression of LMO7. (**A**,**B**) Screening of si-AL137782.1 in A549 and BEAS-2B; (**C**) the mRNA expression of *LMO7* in A549 and BEAS-2B cells was verified by means of RT-qPCR. (**D**,**E**) Western blot was used to verify the protein expression. (**F**,**G**) expression of LMO7 in A549 and BEAS-2B cells was determined by means of immunofluorescence. ** *p* < 0.01, *** *p* < 0.001, ns means no significant difference.

**Figure 7 ijms-24-13904-f007:**
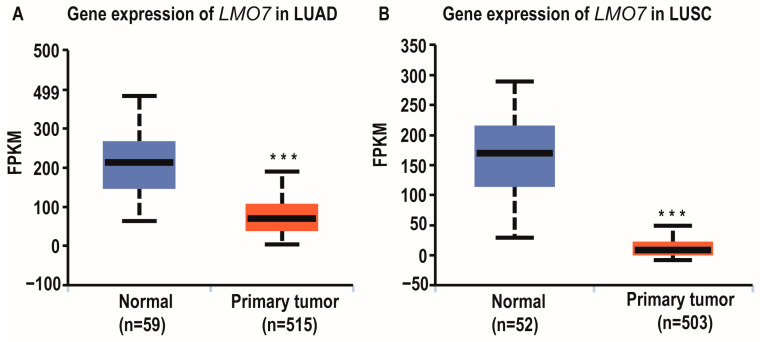
Expression and relative survival rate of lncRNA *LMO7* in lung cancer groups. (**A**) Ex pression of *LMO7* in LUAD; (**B**) expression of *LMO7* in LUSC. *** *p* < 0.001.

**Figure 8 ijms-24-13904-f008:**
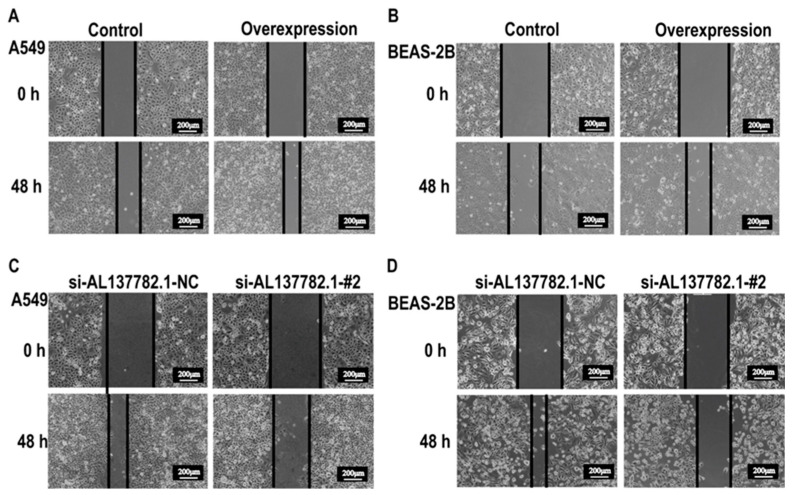
AL137782.1 promotes the migration of lung epithelial cells. (**A**,**B**) Overexpression of AL137782.1 promoted the migration of A549 and BEAS-2B cells; (**C**,**D**) knocking down AL137782.1 suppressed the migration of A549 and BEAS-2B cells.

**Figure 9 ijms-24-13904-f009:**
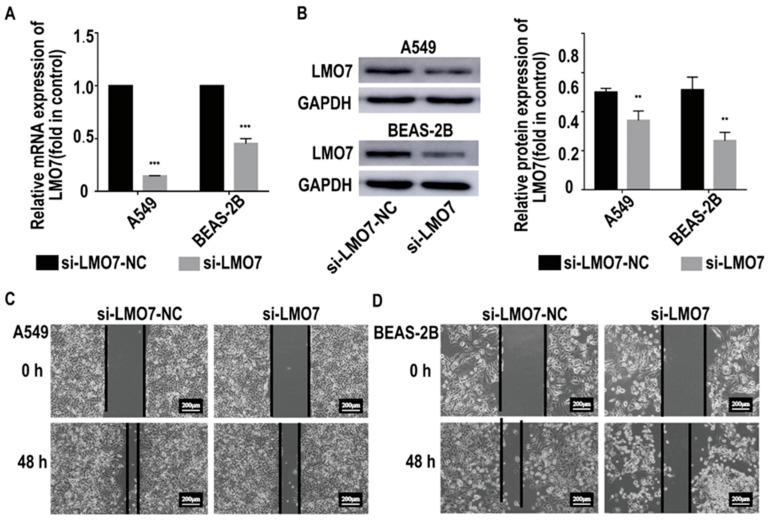
Knocking down LMO7 suppresses the migration of lung epithelial cell while overexpressing the expression of AL137782.1. (**A**) The mRNA expression of *LMO7* in *A549* and BEAS-2B cells were verified by RT-qPCR; (**B**) Western blot was used to verify the protein expression; (**C**,**D**) knocking down *LMO7* suppressed the migration of A549 and BEAS-2B cells while overexpressing the expression of AL137782.1. ** *p* < 0.01, *** *p* < 0.001.

## Data Availability

The original contributions presented in this study are included in the article/Appendix A.

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
