# Peer review of "Novel Long Non-Coding RNA (lncRNA) Transcript AL137782.1 Promotes the Migration of Normal Lung Epithelial Cells through Positively Regulating LMO7"

_ijms, 2023, doi:10.3390/ijms241813904_

Round 1

Reviewer 1 Report

To begin with, upon first reading the manuscript, I noticed the work done by the team for data collection and for the elaboration of this manuscript.

The presented data are of high quality, innovative as well as clinically very relevant.

Please make changes:

1.    Change structure of your manuscript “The structure should include an Abstract, Keywords, Introduction, Materials and Methods, Results, Discussion, and Conclusions (optional) sections”

Author Response

Reviewer 1.    Change structure of your manuscript “The structure should include an Abstract, Keywords, Introduction, Materials and Methods, Results, Discussion, and Conclusions (optional) sections”

The structure of our paper has been changed. This manuscript including introduction (section 1) , results (section 2), discussion (section 3) and materials and methods (section 4).

  • Some sentences have been edited for clarity and conciseness. And a conclusion has been added into abstract, as “Therefore, our study has provided new insights into the mechanism underlying cell migration in human lung epithelial cells. ” (Highlight in yellow)
  • Some grammar errors have been changed, such as “was, were, has, have”. (Highlight in yellow)
  • The passive sentence has been created from the active sentence, such as “we…”, “we found that…”. (Highlight in yellow)
  • One more reference has been added. (Highlight in yellow)    [18]       Wang Y., Zhang D., Li Y., Wu Y., Ma H., Jiang X., Fu L., Zhang G., Wang H., Liu X., et al., Constructing a novel signature and predicting the immune landscape of colon cancer using N6-methylandenosine-related lncRNAs. Front Genet, 2023. 14: p. 906346

Reviewer 2 Report

The article by Ying Zhang and colleagues entitled “Novel lncRNA transcript AL137782.1 promotes the migration of normal lung epithelial cells through positively regulating LMO7” is an interesting manuscript. The authors analysed the role of lncRNA ENSG00000261553 transcript AL137782.1 in lung epithelial cells exposed 12 to microbes. Excellent is the point that the study includes in-silico work and relevant in-vitro experiments.

This manuscript can be published in the journal “International Journal of Molecular Sciences” after minor revisions.

My concerns are:

- The “material and methods” part is poorly. The authors have to improve this part significantly. City and countries must be added to all companies. Sometimes even the company is not mentioned for used reagents and/or equipment. Furthermore, the composition of the used buffers/percentage of solutions must be indicated.

- The authors have to check the manuscript in regard to the use of the English language. Some significant improvements are necessary.

- Some statements like “…. we set up empty vector control group and 124 AL137782.1 overexpression group….” are unscientific and must be improved.

- It is neither nice nor necessary to read “Hu et al.. ….” etc. if somebody is interested in the name of the first author (s)he will find this information in the Reference list. The authors have to rephrase all these parts

The authors have to check the manuscript in regard to the use of the English language. Some significant improvements are necessary.

Author Response

1. The “material and methods” part is poorly. The authors have to improve this part significantly. City and countries must be added to all companies. Sometimes even the company is not mentioned for used reagents and/or equipment. Furthermore, the composition of the used buffers/percentage of solutions must be indicated.

The companies/countries of solutions and kits have been added. (Highlight in yellow)

2. The authors have to check the manuscript in regard to the use of the English language. Some significant improvements are necessary.

  • Some sentences have been edited for clarity and conciseness. And a conclusion has been added into abstract, as “Therefore, our study has provided new insights into the mechanism underlying cell migration in human lung epithelial cells. ” (Highlight in yellow)
  • Some grammar errors have been changed, such as “was, were, has, have”. (Highlight in yellow)
  • The passive sentence has been created from the active sentence, such as “we…”, “we found that…”. (Highlight in yellow)
  • One more reference has been added. (Highlight in yellow)    [18]       Wang Y., Zhang D., Li Y., Wu Y., Ma H., Jiang X., Fu L., Zhang G., Wang H., Liu X., et al., Constructing a novel signature and predicting the immune landscape of colon cancer using N6-methylandenosine-related lncRNAs. Front Genet, 2023. 14: p. 906346

3. Some statements like “…. we set up empty vector control group and 124 AL137782.1 overexpression group….” are unscientific and must be improved.

Before change: To analyze the function of AL137782.1, we set up empty vector control group and AL137782.1 overexpression group, extracted total RNA of A549 cells for transcriptome sequencing.

After change: To analyze the function of AL137782.1, control cell lines and AL137782.1 overexpression cell lines were established. Then total RNA was extracted for transcriptome sequencing. (Highlight in yellow)

4. It is neither nice nor necessary to read “Hu et al.. ….” etc. if somebody is interested in the name of the first author (s)he will find this information in the Reference list. The authors have to rephrase all these parts

Before change: In the study of how MRTFs and SRF drive cell-specific transcription, Hu et al found that LMO7 is a cell-specific regulator of MRTFs, which can alleviate actin-mediated inhibition under the synergistic effect of Rho GTPase, thereby regulates the cell-specific activation of the Rho-MRTF-SRF pathway and plays an important role in breast cancer cell migration.

After change: In the investigation of the mechanisms underlying cell-specific transcription driven by MRTFs and SRF, it has been found that LMO7 acts as a cell-specific regulator of MRTFs. Under the synergistic influence of Rho GTPases, LMO7 might alleviate the inhibitory effect mediated by actin.  Highlight in yellow
